# Comparative efficacy of cisternal drainage versus external ventricular drainage in severe aneurysmal subarachnoid hemorrhage

Mahamat Hamid Mahamat [1,2,3,4,5], Tuo Li[2,3,4,5], Jun Liu[2,3,4,5], Shusheng Zhang[2,3,4,5], Ye Miao[2,3,4,5], Zhongzhen Li[2,3,4,5], Yadan Li[2,3,4,5], Hua Yan[2,3,4,5]*, Guobin Zhang[2,3,4,5]*, Xiaoguang Tong[1,2,3,4,5]*

**1** Nankai University, The school of Medicine; 94 weijin Rd, Tianjin, China, **2** Department of Neurosurgery, Tianjin Huanhu Hospital, Tianjin, China, **3** Brain Trauma, Neurosurgical Critical Care Center Huanhu Hospital, Tianjin, China, **4** Laboratory of Micro neurosurgery, Tianjin Neurosurgical Institute, Tianjin, China, **5** Tianjin Key Laboratory of Cerebral Vascular and Neural Degenerative Diseases, Tianjin, China

* yanhua20042007@sina.com (HY); cntjzgb@hotmail.com (GZ); tongxiaoguanghhh@163.com (XT)

## Abstract

### Background

The cerebrovascular emergency known as aneurysmal subarachnoid hemorrhage (aSAH) is potentially fatal. Although external ventricular drainage (EVD) is the gold standard for monitoring intracranial pressure (ICP), cisternal drainage (CD) should be considered as a supplementary strategy due to the limited effectiveness of EVD in removing inflammatory mediators and preventing vascular damage.

### Objective

To compare ICP monitoring accuracy, cerebrospinal fluid (CSF) clearance, and clinical outcomes between EVD and CD in severe aSAH patients.

### Methods

A retrospective study enrolled 47 Hunt-Hess IV–V grade aSAH patients, divided into EVD (n = 23) and EVD + CD (n = 24) groups. Daily ICP values (days 1/3/5/7), CSF biomarkers (cell count, protein, Interleukin-6, Interleukin-8, Tumor Necrosis Factor-α, Endothelin-1, Monocyte Chemoattractant Protein-1, Vascular Cell Adhesion Molecule-1), hospitalization duration, and 6-month Glasgow Outcome Scale (GOS) were analyzed.

### Results

ICP values showed no significant difference between the two groups ($p > 0.05$). EVD + CD group exhibited higher CSF cell count (394.68 ± 91.32 vs. 320.40 ± 75.49), protein (16.17 ± 2.27 vs. 13.74 ± 2.94 g/L), and cytokines (IL-6/IL-8, $p < 0.05$) on day 1,

**Data availability statement:** All relevant data are within the paper and its Supporting Information files.

**Funding:** This research was funded by Tianjin Municipal Education Commission, research project granted number 2024KJ248.

**Competing interests:** The authors have declared that no competing interests exist.

but significantly lower levels by day 7 ($p < 0.05$). Vascular injury factors (ET-1/MCP-1/VCAM-1) were reduced in the EVD + CD group by day 7 ($p < 0.05$). Hospitalization duration was 22.8% shorter in the EVD + CD group (27 vs. 35 days, $p = 0.030$); while 6-month GOS showed no difference (2.52 vs. 2.57, $p = 0.148$).

## Conclusion

Compared with EVD, CD could not only provide accurate ICP readings and trend data but also enhance inflammatory clearance, reduce hospitalization time, and mitigate vascular injury in severe aSAH, warranting further validation in larger cohorts.

## Introduction

Aneurysmal subarachnoid hemorrhage (aSAH) is a potentially life-threatening neurological event resulting from the rupture of an intracranial aneurysm, leading to the accumulation of blood in the subarachnoid space. The sudden increase in intracranial pressure (ICP) and irritation of the brain tissue and meninges can result in significant neurological deficits and long-term complications, highlighting the importance of prompt diagnosis and intervention [1]. The global incidence of aSAH is estimated to be between 6 and 9 per 100,000 person-years [2]. Mortality rates for aSAH are high, with up to 50% of cases resulting in death, and a significant proportion of survivors experiencing long-term functional impairment [3].The initial hemorrhage introduces blood directly into the subarachnoid space, triggering a cascade of secondary insults including acute intracranial hypertension, neuroinflammation, and delayed cerebral ischemia (DCI) [4]. These processes are largely driven by the prolonged presence of blood breakdown products, such as oxyhemoglobin, and a consequent surge in pro-inflammatory cytokines within the cerebrospinal fluid (CSF) [5,6].

The cornerstone of managing elevated ICP and hydrocephalus in severe aSAH is external ventricular drainage (EVD). As the gold standard for ICP monitoring, EVD provides a critical therapeutic window for controlling cerebral perfusion pressure [7]. However, its efficacy in evacuating the widespread inflammatory and vasoactive substances sedimented in the basal cisterns—the epicenter of secondary brain injury—is inherently limited by the ventricular anatomy and CSF flow dynamics.

Cisternal drainage (CD) has been proposed as a complementary strategy to address this limitation. By placing a catheter directly into the optico-carotid and pre-pontine cisterns during surgical clipping, CD targets the primary reservoir of hemorrhagic blood. Previous studies have suggested that CD may improve outcomes by reducing the risk of cerebral vasospasm, potentially through more effective clearance of subarachnoid clots [8,9]. Nevertheless, the existing literature lacks a direct, simultaneous comparison of the biochemical profiles of CSF drained via EVD versus CD in the same patient cohort. Key questions remain unanswered: Does CD indeed provide a more efficient clearance of inflammatory mediators and vascular injury markers? And critically, can it deliver ICP monitoring data that is as reliable as the established EVD gold standard?

To bridge this knowledge gap, we conducted a comparative study in patients with severe (Hunt-Hess grade IV-V) aSAH. Our primary objectives were twofold: first, to validate the accuracy of ICP monitoring via CD against simultaneous EVD measurements; and second, to quantitatively assess and compare the temporal dynamics of CSF purification, including the clearance of cells, proteins, key inflammatory cytokines, and pivotal vascular injury factors between the two drainage methods. This investigation aims to provide a comprehensive pathophysiological rationale for the potential integration of CD into the multimodal management strategy for severe aSAH.

## Materials and methods

### Study design and population

A single-center retrospective cohort study included 47 patients with Hunt-Hess IV–V grade aSAH treated between October 1st 2021 and October 31st 2023. Inclusion criteria: [1] Patients aged 18 years or older; [2] Patients with Hunt-Hess grade IV–V on admission; [3] Patients showing subarachnoid hemorrhage on cranial CT scan at admission; [4] Intracranial aneurysm confirmed by CTA, MRA, or DSA during admission; [5] EVD completed preoperatively; [6] Patients requiring craniotomy for aneurysm clipping combined with cisternal opening and catheter drainage. Based on Hunt-Hess grading, patients were further categorized into either the EVD group or the EVD + CD group. The preoperative and postoperative images are shown in Figs 1 and 2. Exclusion criteria: [1] No intracranial aneurysm detected on imaging studies; [2] Patients with confirmed intracranial aneurysm who declined treatment (voluntarily discharged) or opted for endovascular intervention; [3] History of hydrocephalus or coexisting intracranial pathologies (e.g., cerebral trauma, brain tumor, cerebrovascular malformation); [4] Patients who did not undergo surgical clipping for aneurysm treatment; [5] Age > 80 years, late-stage cerebral herniation, or multi-organ dysfunction (cardiac, pulmonary, hepatic, renal) contraindicating craniotomy.

### Interventions

Patients with aSAH were categorized into either the EVD group or the EVD + CD group. EVD group: A catheter was placed in the lateral ventricle via Kocher point before craniotomy. EVD + CD group: Before craniotomy, EVD was completed, and

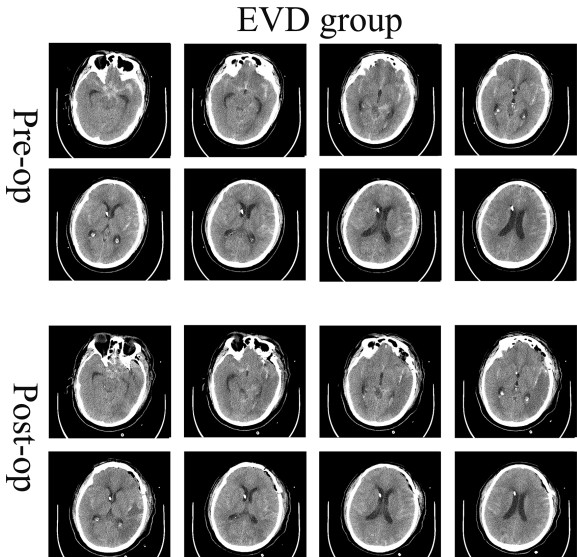

**Fig 1. Postoperative imaging manifestations of patients in the EVD group.** EVD group: A catheter was placed in the lateral ventricle via Kocher point before craniotomy. Reprinted from Tianjin huanhu hospital under a CC BY license, with permission from Plos one, original copyright 2025.

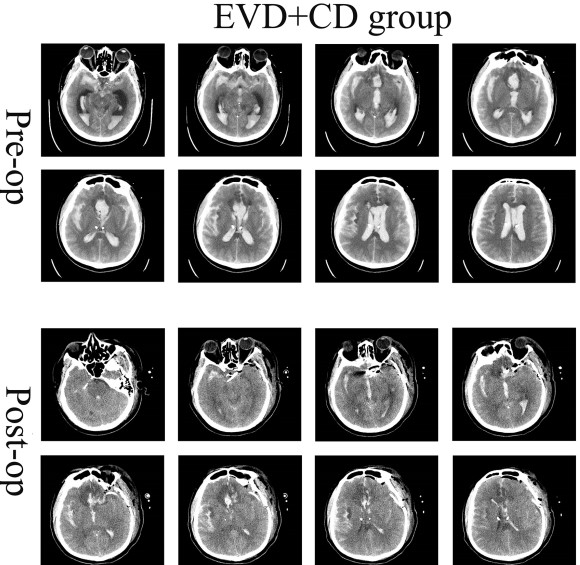

**Fig 2. Postoperative imaging manifestations of patients in the EVD + CD group.** EVD + CD group: EVD was finished before craniotomy, and a catheter was inserted in the prepontine cistern after the optico-carotid cistern. Reprinted from Tianjin huanhu hospital under a CC BY license, with permission from Plos one, original copyright 2025.

during craniotomy, the optico-carotid cistern and Liliequist's membrane were opened, lamina terminalis fenestration was simultaneously performed and a catheter was positioned in the prepontine cistern. After craniotomy, both groups received standardized postoperative care, including nimodipine infusion and ICP-targeted CSF drainage (100 mL/day).

**Cisternal drain technique**

The surgical method for accessing the basal cisterns, known as cisternostomy, has been previously outlined for aSAH [10,11]. A Mayfield clamp is used to hold the patient's head in place as it is stretched and rotated 30 degrees to the other side. The orbital roof is flattened, and the sphenoid ridge is drilled as part of a standard pterional craniotomy. Near the basal dura, a curvilinear frontotemporal durotomy is performed. The olfactory nerve is detected first, then the optic nerve medially, using a lateral subfrontal approach. This makes it easier to open the optico-carotid cistern and the Liliequist membrane, which makes it possible to see the posterior circulation through this aperture. Direct connection with the third ventricle is made possible by more medial access to the lamina terminalis posteriorly and the interventricular space anteriorly. Lamina terminalis fenestration was simultaneously performed, thereby improving CSF flow dynamics. Typically, a soft drainage catheter is then placed in the optico-carotid cistern, aiming to reduce the risk of cerebral vasospasm, alleviate meningeal irritation caused by bloody CSF and monitor dynamic changes in intracranial pressure. Fig 3 illustrates the relevant anatomy of this procedure.

**Data collection**

Epidemiological data, clinical and radiological characteristics were collected through a retrospective review of the electronic medical records.

Hunt-Hess IV–V were categorized as high scores. The location and size of the aneurysm were identified using preoperative angio-CT or angio-MR. Data regarding the surgical procedure, including surgical treatment and the placement of EVD or CD, were gathered. Additionally, information on post-procedural complications was collected.

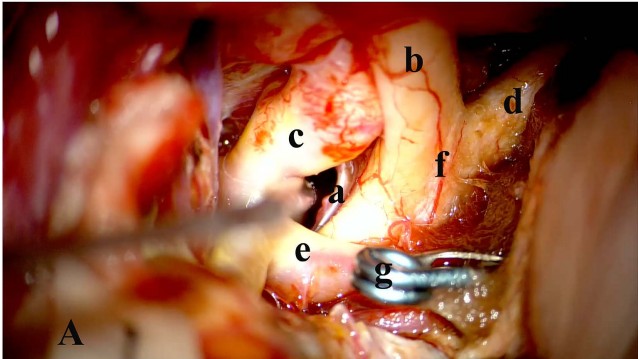
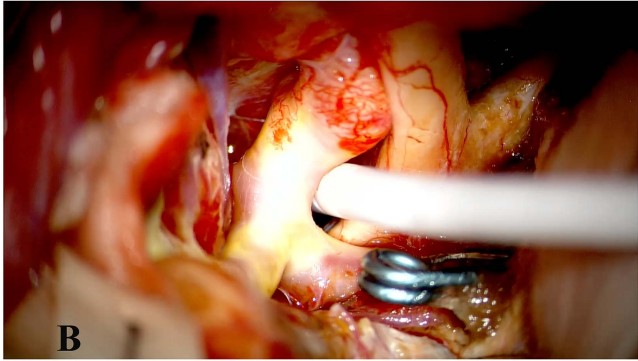

**Fig 3. Procedure for cisternal drain placement. a.** Optico-carotid Cistern, **b.** Optic nerve ipsilateral, **c.** Internal Carotid Artery (ICA), **d.** Optic nerve contralateral, **e.** Anterior Cerebral Artery (ACA), **f.** Optic-Chiasm, **g.**Clipping. The image acquired following the opening of the optico-carotid cistern with the internal carotid artery (ICA), optic nerve (ON), and Liliequist's membrane is depicted in this figure. Observe how the catheter is positioned between the internal carotid artery (ICA) and the optic nerve (ON), traveling through Liliequist's membrane and into the prepontine cistern. Reprinted from Tianjin huanhu hospital under a CC BY license, with permission from Plos one, original copyright 2025.

The time from the incidence of aSAH and the most recent documented neurosurgical or neuroradiological consultation was defined as the follow-up period. The Glasgow Outcome Scale (GOS) was used to measure clinical outcomes at the time of hospital release and six months later.

ICP Monitoring: The comparison of differences in intracranial pressure monitoring between the two drainage methods is primarily conducted in the EVD + CD group. Since 3 patients passed away within seven days, only 21 sets of data remain in the end. In order to determine whether there is a difference in intracranial pressure monitored by the two drainage methods, we continuously observed the dynamic differences in intracranial pressure through external ventricular drainage and cisternal drainage on the 1st, 3rd, 5th, and 7th days post-surgery. Each day, both methods were used to dynamically measure intracranial pressure in 10 sets, and then the average value was taken as the patient's intracranial pressure for that day and recorded.

CSF Analysis: The ultimately obtained data consisted of 20 sets in the EVD group and 19 sets in the EVD + CD group. CSF samples were analyzed for cell count, proteins, Interleukin-6 (IL-6), Interleukin-8 (IL-8), Tumor Necrosis Factor-α (TNF-α), Endothelin-1 (ET-1), Monocyte Chemoattractant Protein-1 (MCP-1), and Vascular Cell Adhesion Molecule-1 (VCAM-1). The cell count and protein content of cerebrospinal fluid (CSF) were analyzed after collecting the CSF and sending it to the laboratory at Tianjin Huanhu Hospital for testing. For the detection of inflammatory factors, after successfully collecting the CSF, it was centrifuged at 3000 rpm for 10 minutes in a 4°C centrifuge, and then frozen at −80°C until our next step of testing and analysis. We purchased enzyme-linked immunosorbent assay (ELISA) kits for IL-6 (Torey Fuji, Japan), IL-8 (Torey Fuji, Japan), TNF-α (Ohtsuka Pharm, Japan), ET-1 (Xinfan Biotechnology, Shanghai, China), MCP-1

(R&D Systems USA), and VCAM-1 (R&D Systems USA) for detection. The assay procedure was strictly adhered to according to the manufacturer's instruction.

The collection and organization of clinical data, follow-up data, and experimental data were completed on January 15, 2025, for use in subsequent analysis.

### Ethics statement

This human subject study was approved by the Ethics Committees of Tianjin Huanhu Hospital (2020−75). All participants (or legal representatives) were informed of the study protocol and signed the informed consent form. All methods were carried out according to relevant guidelines and regulations.

### Statistical analysis

This experiment used IBM SPSS Statistics 26 and GraphPad Prism 9 software for plotting and statistical analysis. As for the ICP data analysis, the Shapiro-Wilk test was used to check whether the data followed a normal distribution, the F-test was used for testing the homogeneity of variances, and the Bonferroni correction was applied to adjust the significance level to avoid the accumulation of false positive errors. Subsequently, a paired t-test was employed to observe whether there were significant differences between the two drainage methods in terms of intracranial pressure monitoring. As for the CSF Analysis, the Kolmogorov-Smirnov test is used to check whether the data follows a normal distribution, the F-test is used for testing the homogeneity of variances, and Tukey's post hoc test is used for post hoc analysis of the data. Non-continuous variables (such as cerebrospinal fluid cell count, cerebrospinal fluid protein levels, IL-6, IL-8, TNF-α, ET-1, MCP-1, and changes in VCAM-1) are analyzed using repeated measures ANOVA. As for the clinical data, Binary logistic regression analysis to adjust for potential confounders (such as gender, age, GCS, Hunt-Hess, hospitalization and GOS). Data are presented as mean ± standard deviation, and a $p$-value less than 0.05 is considered to indicate a statistically significant difference.

## Results

### ICP monitoring

In the EVD + CD group, both EVD and CD were used for ICP monitoring. EVD provided continuous pressure monitoring, while CD offered continuous CSF drainage with intermittent pressure measurements. By using a hydraulic coupling, intracranial pressure monitoring through the ventricles and cisterns, we simultaneously collected intracranial pressure. In patients who had both EVD and CD, we tracked the dynamic changes in ICP following surgery. To determine whether there were any differences between the two measuring techniques, we compared the intracranial pressure readings obtained using the two procedures.

Through Fig 4, we can observe that there is no significant statistical difference in ICP measured by the two methods on postoperative days 1, 3, 5, and 7. This indicates that compared to traditional EVD for pressure measurement, CD can accurately monitor changes in ICP. Although the positions of the drainage tube tips differ between the ventricle and cistern, there is no significant difference in the ICP measured by the two drainage methods. As a result, in future clinical practice, if only CD is available, it can provide clinicians with trustworthy ICP data while also draining CSF, producing nearly the same impact as EVD.

### Routine and biochemical changes in CSF

CSF was collected from the ventricular drainage tube in the EVD group and the cistern drainage tube in the EVD + CD group. On the 1st, 3rd, 5th, and 7th days following surgery, we took CSF samples, which we then forwarded to the hospital's lab for routine and biochemical analysis. From Fig 5A and 5B, we could observe the changes in cell count and protein

## ICP Monitoring in EVD+CD group

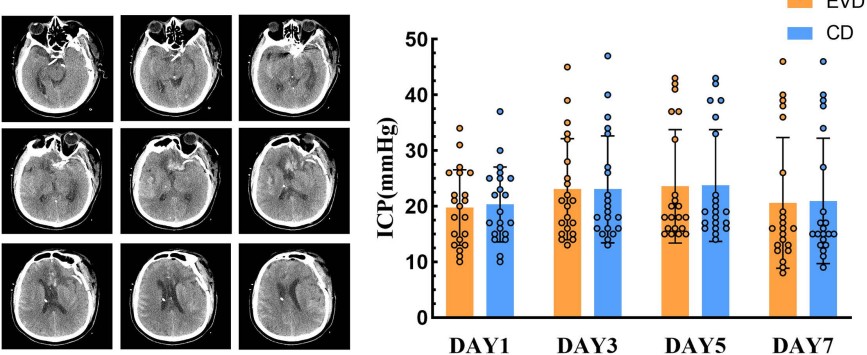

**Fig 4. Postoperative intracranial pressure monitoring.** ICP Monitoring: While cisternal drainage enabled continuous CSF drainage with sporadic pressure readings, external ventricular drainage offered constant pressure monitoring. ICP on postoperative days 1, 3, 5, and 7, as determined by the two approaches, does not significantly differ statistically. Reprinted from Tianjin huanhu hospital under a CC BY license, with permission from Plos one, original copyright 2025.

levels in the CSF. It was evident that in the two groups, the highest levels of cell count and protein levels emerged on the first postoperative day, which gradually decreased over time as the CSF was drained and clarified. Additionally, we discovered that, with some statistically significant differences, the CSF from the EVD+CD group had a considerably greater cell count and protein content on the first postoperative day than that of the EVD group. This indicated that the cistern drainage effectively drained cells and proteins in the early stage, leading to a significant decrease in cell count and protein content in the EVD+CD by the seventh postoperative day, which were notably lower than those in the EVD group. This suggested that within the same timeframe, the drainage in the EVD+CD group could be more effective, thereby reducing the total cell count and protein levels in the CSF, which facilitated the clarification of the CSF.

### Changes in inflammatory factors in cerebrospinal fluid

Additionally, we investigated how the inflammatory components in CSF changed over time following surgery. From Fig 5D-5F, it was evident that both groups had the highest levels of IL-6, IL-8, and TNF-α in the CSF on the first postoperative day, and the levels of inflammatory components in the CSF significantly dropped as the follow-up period went on and the CSF was progressively drained and cleared. Moreover, we found that on the first postoperative day, the levels of IL-6 and IL-8 in the CSF drained from the CD tubes were significantly higher than those in the EVD tubes, with some statistically significant differences. This indicated that the cisternal drainage had a better CSF clearance effect, leading to a situation where, by the seventh postoperative day, the levels of inflammatory factors in the CSF drained from the EVD+CD group were significantly lower than those in the EVD group.

### Changes in markers of vascular endothelial cell injury

Additionally, we tracked how the markers of vascular endothelial injury changed in CSF at various intervals after surgery. From Fig 5G-5I, we could see that the levels of ET-1 in the CSF of both groups were highest on the first postoperative day, while MCP-1 and VCAM-1 peaked on the 3rd and 5th postoperative days, respectively. As time progressed, the levels of vascular injury markers in the CSF gradually decreased. Moreover, we observed that on the first postoperative day, although there was no significant statistical difference, the levels of ET-1, MCP-1, and VCAM-1 in the CSF from the EVD+CD group were slightly higher than those in the EVD group. On the third postoperative day, the levels of MCP-1 and VCAM-1 in the CSF from the EVD+CD were significantly higher than those in the EVD group. And by the seventh

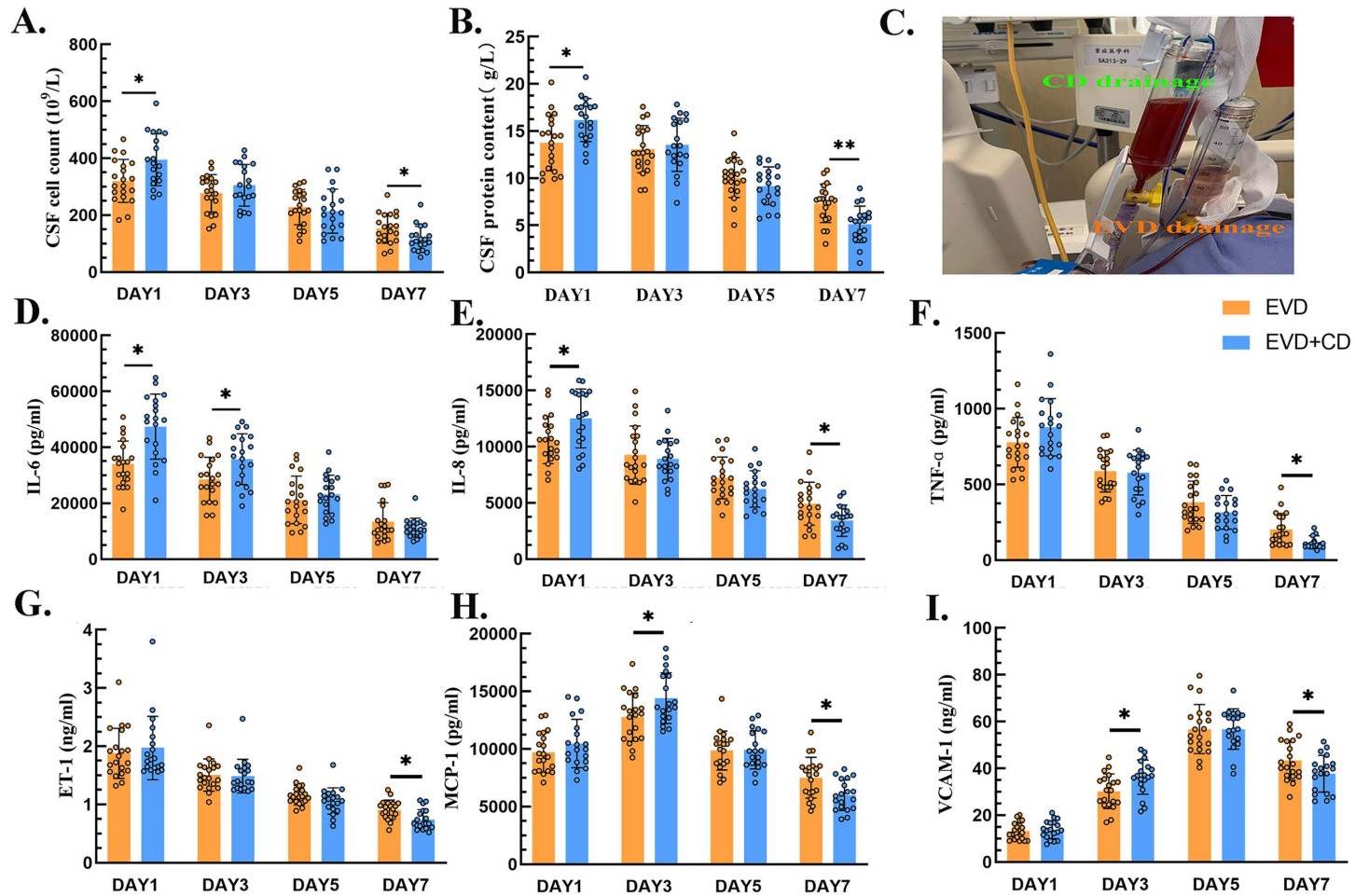

**Fig 5. Changes in cerebrospinal fluid characteristics and biomarkers.** A and B: cell count and protein content of CSF. D-F: Changes in inflammatory factors. G-I: Changes of vascular endothelial cell injury factors. C: Different characteristics of CSF in the two groups. From those pictures, we could infer that cisternal drainage could enhance hemorrhagic/inflammatory CSF clearance. (*$p < 0.05$, **$p < 0.01$). Reprinted from Tianjin huanhu hospital under a CC BY license, with permission from Plos one, original copyright 2025.

postoperative day, the levels of ET-1, MCP-1, and VCAM-1 in the CSF from the EVD + CD group were significantly lower than those in the EVD group.

### Clinical outcomes

From Table 1, there were no significant differences in age, gender, GCS scores, and Hunt-Hess scores between the two groups of patients before surgery. Additionally, there were no statistical differences in GOS scores at 6 months post-surgery, which may be related to the fact that we selected only patients with severe subarachnoid hemorrhage, or it could be due to our sample size being insufficient. However, we observed that the hospitalization time in the treatment group with EVD + CD drainage was significantly shorter than that in the EVD group.

### Discussion

This study aimed to address a significant gap in the management of severe aSAH by directly comparing the efficacy of EVD combined with CD against EVD alone. Specifically, we sought to evaluate whether the adjunctive use of CD could

**Table 1. Clinical data of aneurysmal subarachnoid hemorrhage (aSAH) (47 patients).**

|  | EVD group (n = 23) | EVD + CD group(n = 24) | *P* value |
|---|---|---|---|
| age | 59(41-82) | 58(39-80) | 0.610 |
| male | 12(52.17%) | 10(41.67%) | 0.295 |
| Aneurysm location |  |  | 0.354 |
| anterior communicating artery | 6(26.09%) | 6(25%) |  |
| Middle cerebral artery | 8(34.78%) | 6(25%) |  |
| Anterior cerebral artery | 4(17.39%) | 6(25%) |  |
| posterior communicating artery | 3(13.04%) | 4(16.67%) |  |
| Internal carotid artery | 2(8.70%) | 2(8.33%) |  |
| GCS | 5.87( 3 –9 ) | 5.79( 3 –9 ) | 0.637 |
| Hunt-Hess | 4.52( 4 –5 ) | 4.54( 4 –5 ) | 0.972 |
| Hospitalization(day) | 35(8-68) | 27(4-50) | 0.030 |
| GOS(6 months) | 2.57( 1 –4 ) | 2.52( 1 –4 ) | 0.148 |

provide accurate intracranial pressure measurements and enhance the clearance of neurotoxic blood breakdown products and inflammatory mediators from the cerebrospinal fluid, beyond what is achievable with standard EVD, and to assess its impact on clinical parameters.

Through ICP monitoring experiments, we found that CD can provide accurate intracranial pressure measurements. Meanwhile, in the EVD + CD group, the characteristics of the CSF drained by the two methods were different, as shown in Fig 5C: cisternal drainage effluent contained substantially greater blood components and particulate matter, resulting in pronounced visual turbidity. During the process of red blood cell lysis, a large number of harmful substances are generated, which in turn cause secondary brain injury [12]. Secondary brain injury mainly includes pathological processes such as oxidative stress and inflammatory response. Among the pro-inflammatory factors detected, IL-6 contributes to the occurrence of delayed cerebral vasospasm by promoting endothelial cell damage and the release of vasoconstrictors such as ET-1 [13,14]. IL-8, as a potent chemokine, attracts neutrophils to infiltrate the hemorrhagic site, releasing reactive oxygen species (ROS) and proteases, which aggravate tissue damage [15,16]. In the early stages of SAH, TNF-α is released by microglia and endothelial cells, activating the NF-κB pathway and driving a cascade of inflammatory factors such as IL-6 and IL-8 [17–19]. TNF-α promotes platelet aggregation and microthrombus formation, exacerbating DCI [20,21]. From the biochemical testing, routine testing, and inflammatory factors detection, the levels of cell count, protein count, IL-8, and TNF-α in the EVD + CD group were significantly lower than those in the EVD group on the seventh postoperative day. It was reasonable to speculate that cisternal drainage offers superior CSF clearance efficacy. A higher rate and greater volume of bloody CSF drainage are associated with reduced severity of secondary brain injury.

As reported that cerebral vasospasm is a frequent complication of aSAH, and leukocyte-endothelial cell interactions play a significant role in the pathophysiology of cerebral vasospasm [22], we monitored the changes of ET-1, MCP-1, and VCAM-1, which could play important roles in vasospasm. ET-1, a potent bioactive peptide primarily synthesized by vascular endothelial cells, is among the strongest known vasoconstrictors, inducing potent constriction throughout the vascular system. After SAH, when blood enters the subarachnoid space, it stimulates the release of ET-1 from vascular endothelial cells and other sources [23]. Additionally, the pathophysiological processes, such as cerebral vasospasm that occur after hemorrhage, can further increase the synthesis and release of ET-1. At the same time, ET-1 can induce cerebral vasospasm again, worsening the degree of vasospasm and subsequently affecting the patient's prognosis [23–25]. Therefore, the level of ET-1 in the cerebrospinal fluid is closely related to the severity of the condition and prognosis after subarachnoid hemorrhage [26]. MCP-1 regulates endothelial cell functions (adhesion molecule expression, vascular permeability) and drives CNS microglial activation and neuroinflammation [27,28]. Post-SAH hematoma absorption activates

astrocytes and microglia, which elevate MCP-1 to recruit monocytes and polarize macrophages toward pro-inflammatory phenotypes, amplifying ROS release and oxidative stress damage [29,30]. Concurrently, MCP-1 stimulates ET-1 synthesis, amplifying vasoconstriction [31]. VCAM-1 is predominantly expressed on activated endothelial cells and vascular smooth muscle cells, mediating leukocyte adhesion, migration, and activation [32,33]. Post-SAH, inflammatory cytokines such as TNF-α induce endothelial VCAM-1 expression [34]. Additionally, VCAM-1 promotes smooth muscle cell migration and proliferation, aggravating vasospasm and microcirculatory dysfunction [32,35]. As key post-SAH inflammatory mediators, ET-1, MCP-1, and VCAM-1 drive neurological injury stages (neuroinflammation, vasospasm, oxidative damage) via multi-target pathways. Post-SAH elevation of inflammatory and vascular injury markers reflects intracranial oxidative stress and hyperinflammation, culminating in endothelial damage and vasospasm. Therefore, adequate postoperative CSF drainage becomes critically important. With more effective CSF clearance, ET-1, MCP-1, and VCAM-1 significantly decreased, then we could infer that with more ET-1, MCP-1, and VCAM-1 being drained out, the likelihood of vasospasm would decrease. Nevertheless, definitive confirmation requires subsequent vascular imaging, and delayed infarcts are not assessed by the criteria suggested by Vergouwen [36], highlighting a critical component for methodological refinement in upcoming clinical trials.

Collectively, a key finding of our study was the superior CSF clearance efficacy of the EVD + CD approach, which, despite initially higher levels of cellular debris, proteins, and inflammatory cytokines in CD effluent on day 1, led to markedly cleaner CSF profiles by day 7. This supports CD's role as a "source control" mechanism, directly targeting the basal cisterns where inflammatory substances accumulate, consistent with the physiological model of cisterns as terminal CSF reservoirs and extending the observations of Vandenbulcke [37] by providing direct biochemical evidence. Furthermore, the significant reduction in vascular injury markers (ET-1, MCP-1, VCAM-1) by day 7 suggests a mechanistic link whereby effective subarachnoid evacuation disrupts the cycle of endothelial activation and neuroinflammation, potentially modifying the pathophysiology of vasospasm which had been reported in previous clinical experiment [38,39].

Following this, we statistically analyzed the differences between the two patient cohorts. The primary aim of this study was to evaluate the long-term functional recovery and its influencing factors specifically in aSAH patients. Both groups showed no significant differences in preoperative age, sex, Glasgow Coma Scale (GCS) scores, or Hunt-Hess grades. Additionally, the 6-month Glasgow Outcome Scale (GOS) scores did not differ statistically. However, the EVD + CD group exhibited a significantly shorter hospital stay compared to the EVD group. The choice between EVD and EVD + CD at our institution was indeed influenced by a combination of factors, including the treating physician's preference and their interpretation of the patient's clinical and radiological status, rather than a strict randomized protocol. However, to address the potential bias introduced by this non-randomized allocation, we performed binary logistic regression analysis to adjust for potential confounders (such as gender, age, GCS, Hunt-Hess, hospitalization and GOS). Even after adjustment, the association between hospitalization and the two different groups remained significant.

However, several limitations of our study must be acknowledged. Firstly, the single-center, retrospective design and relatively small sample size inherently limit the generalizability of our findings and introduce the potential for selection bias. The non-randomized allocation of patients to treatment groups, though adjusted for statistically, remains a source of potential confounding. Secondly, a key methodological limitation, as rightly pointed out in peer review, was that our definition of delayed cerebral infarction did not adhere to the rigorous imaging-based gold standard proposed by Vergouwen [36]. This may have impaired our ability to precisely detect and attribute infarcts to vasospasm, potentially underestimating the true impact of CD on this critical outcome. Furthermore, previous studies have demonstrated that lumbar drainage can also achieve favorable outcomes in the treatment of subarachnoid hemorrhage [40]. Future investigations could compare the relative efficacy of lumbar drainage versus cisternal drainage.

To address these unresolved issues, future research must prioritize a multi-center, prospective, randomized controlled trial design. Such a study should incorporate strict, protocol-driven imaging at defined intervals (24–48 hours and 3–6 weeks post-operatively) to definitively diagnose delayed infarcts. Additionally, correlating CSF biomarker levels with

radiographic evidence of vasospasm on serial angiography would strengthen the causal inference between CD, inflammatory clearance, and vascular protection.

## Conclusion

The insights gained from this investigation lead us to propose a new hypothesis: the primary benefit of cisternal drainage may extend beyond simple ICP management to a fundamental "chemical clearance" of the subarachnoid space. By more effectively removing the soup of pro-inflammatory and vasoactive substances at their source, CD may create a more favorable biochemical environment for vascular and neural recovery. Based on this, we recommend that subsequent studies not only focus on traditional clinical outcomes but also employ advanced multimodal monitoring and high-throughput biomarker profiling to comprehensively map the biochemical and cellular responses elicited by different drainage strategies. This deeper understanding will be crucial for optimizing patient selection and refining the integration of cisternal drainage into the complex management algorithm of severe aSAH.

## Supporting information

**S1 Table. Clinical data of aneurysmal subarachnoid hemorrhage (aSAH).** This file contains the clinical information of all enrolled patients, including age, gender, aneurysm location, and so on.
(XLSX)

**S1 Fig. Postoperative intracranial pressure monitoring.** This document records the dynamic intracranial pressure data from the two groups of patients at various time points following surgery, which was presented in Fig 4.
(XLSX)

**S2 Fig. Changes in cerebrospinal fluid characteristics and biomarkers.** This document comprises the serial measurements of routine CSF analysis, biochemical profiles, and all ELISA-based markers across postoperative time points in the two cohorts, which was presented in Fig 5.
(RAR)

## Acknowledgments

We would like to thank all the participants in the study.

## Author contributions

**Conceptualization:** Mahamat Hamid Mahamat, Tuo Li, Guobin Zhang, Xiaoguang Tong.

**Data curation:** Mahamat Hamid Mahamat, Tuo Li.

**Formal analysis:** Mahamat Hamid Mahamat, Tuo Li, Hua Yan.

**Funding acquisition:** Xiaoguang Tong.

**Investigation:** Tuo Li, Guobin Zhang.

**Methodology:** Mahamat Hamid Mahamat, Tuo Li, Guobin Zhang.

**Project administration:** Jun liu.

**Resources:** Jun liu, Shusheng Zhang, Ye Miao.

**Software:** Shusheng Zhang, Ye Miao, Zhongzhen Li.

**Supervision:** Zhongzhen Li, Hua Yan.

**Validation:** Yadan Li.

**Visualization:** Yadan Li, Hua Yan, Guobin Zhang, Xiaoguang Tong.

**Writing – original draft:** Mahamat Hamid Mahamat, Tuo Li.

**Writing – review & editing:** Mahamat Hamid Mahamat, Tuo Li.

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
