## [Decision Letter · Decision Letter 0]

7 Oct 2025

Dear Dr. Mahamat,

Thank you for submitting your manuscript to PLOS ONE. After careful consideration, we feel that it has merit but does not fully meet PLOS ONE’s publication criteria as it currently stands. Therefore, we invite you to submit a revised version of the manuscript that addresses the points raised during the review process.

We look forward to receiving your revised manuscript.

Kind regards,

Eric Anthony Sribnick, MD, PhD, FAANS

Academic Editor

PLOS ONE

Journal Requirements:

5. We note that Figure(s) 1 to 5  in your submission contain copyrighted images. All PLOS content is published under the Creative Commons Attribution License (CC BY 4.0), which means that the manuscript, images, and Supporting Information files will be freely available online, and any third party is permitted to access, download, copy, distribute, and use these materials in any way, even commercially, with proper attribution. For more information, see our copyright guidelines: http://journals.plos.org/plosone/s/licenses-and-copyright.

a. You may seek permission from the original copyright holder of Figure(s) 1 to 5 to publish the content specifically under the CC BY 4.0 license.

Additional Editor Comments :

**Academic Editor:**

Please review the comments by the reviewers carefully, please include their entire review in your "response to reviewers," and please address their comments in a point-by-point fashion. I have carefully read your manuscript, their reviews, and I think that addressing their points will improve your study.

Best regards,

Eric Sribnick, MD, PhD

Academic Editor

PLOS One

**Reviewers' comments:**

Reviewer's Responses to Questions

**Comments to the Author**

1. Is the manuscript technically sound, and do the data support the conclusions?

Reviewer #1: Partly

Reviewer #2: Yes

2. Has the statistical analysis been performed appropriately and rigorously?

Reviewer #1: Yes

Reviewer #2: Yes

3. Have the authors made all data underlying the findings in their manuscript fully available?

Reviewer #1: Yes

Reviewer #2: Yes

4. Is the manuscript presented in an intelligible fashion and written in standard English?

Reviewer #1: Yes

Reviewer #2: Yes

Reviewer #1: I thank the editors for the opportunity to review the manuscript (PONE-D-25-33307).

This is a small (31 patients) retrospective study of poor grade aSAH patients. The authors focused on a comparision of CSF biomarkers in patients treated with an external ventricular drain (EVD) only compared to treatment with an EVD and an additional cisternal drain (CD).

What is meant by cisternal drain? The study only included patients in whom ruptured aneurysms of the anterior circulation were secured by surgical clipping (i.e. where the surgical access is through the basal cisterns of the brain. After aneurysm clipping, an additional drain was placed in the basal cisterns (opto-chiasmatic cistern). Fenestration of the lamina terminalis is mentioned but not reported if or in which frequency it was performed.

It is not reported what was the rationale for EVD or EVD+CD. Was this two different time periods, physisicans’ preference, patient criteria?

Several CSF markers of inflammation, blood break down and brain injury were sampled from either the EVDs or the CDs. Patients with CDs had higher release of these proteins in the early phase and lower levels after a week of treatment (i.e. a better clearance of these substances).

The study excludes patients with a postoperative survival of <6 months after aSAH and those with clinical loss to follow-up. It is entirely uncomprehensible why this was done and, of course, this makes the outcome assessment fully invalid.

Delayed infarcts (as assessed by the criteria suggested by Vergouwen et. al - i.e. new infarcts visible on a CT or MRI 3-6 weeks after aSAH not present on a scan performed 24-48h after early aneurysm securing) are not assessed as an endpoint.

Several studies that have assessed basal cistern catheter concepts in the past are not discussed and referenced.

Reviewer #2: This manuscript compares cisternal drainage (CD) with external ventricular drainage (EVD) in severe aneurysmal subarachnoid hemorrhage (aSAH). The study addresses a relevant clinical question, is well-structured, and presents interesting findings regarding inflammatory clearance and hospitalization duration. The conclusions are generally supported by the data. However, several minor points should be addressed before publication.

Minor Comments:

Figures: Please ensure all figures (Figure 1-5) are explicitly cited in the main text at the appropriate points in the Results section (e.g., after the first mention of ICP results, CSF analysis, etc.) to improve readability.

Statistical Methods: A brief justification for the choice of specific statistical tests (e.g., repeated measures ANOVA for some variables) would strengthen the Methods section.

Conclusion: Consider slightly tempering the conclusion to emphasize that these promising results, particularly regarding functional outcomes, require validation in larger, prospective cohorts.

The manuscript presents valuable insights and is recommended for acceptance after these minor revisions.

**Do you want your identity to be public for this peer review?** For information about this choice, including consent withdrawal, please see our Privacy Policy

Reviewer #1: No

Reviewer #2: No

---

## [Author Response · Author response to Decision Letter 1]

29 Oct 2025

We sincerely extend our gratitude to the editorial team and the reviewers for their insightful comments and valuable suggestions provided during the previous round of review. These recommendations have been instrumental in significantly enhancing the scientific rigor and clarity of presentation of our study.

In response to all the reviewers' comments, we have undertaken a comprehensive and meticulous revision of the original manuscript.We firmly believe that these revisions have substantially elevated the methodological robustness, reliability of the results, and overall persuasiveness of the conclusions.

We kindly request that you consider our revised manuscript for the next stage of the review process and await your further decision.

Thank you once again for your time and consideration.

---

## [Editor Report · Decision Letter 1]

5 Nov 2025

Dear Dr. Mahamat,

Thank you for submitting your manuscript to PLOS ONE. After careful consideration, we feel that it has merit but does not fully meet PLOS ONE’s publication criteria as it currently stands. Therefore, we invite you to submit a revised version of the manuscript that addresses the points raised during the review process.

**As requested previously, in your response to reviewers portion, please include BOTH the query raised by the reviewer and your response (i.e., how you addressed the query).**

Thank you,

Academic Editor

We look forward to receiving your revised manuscript.

Kind regards,

Eric Anthony Sribnick, MD, PhD, FAANS

Academic Editor

PLOS ONE

Journal Requirements:

Additional Editor Comments:

As requested previously, in your response to reviewers portion, please include BOTH the query raised by the reviewer and your response (i.e., how you addressed the query).

Thank you,

Eric

---

## [Author Response · Author response to Decision Letter 2]

11 Nov 2025

We sincerely extend our gratitude to the editorial team and the reviewers for their insightful comments and valuable suggestions provided during the previous round of review. These recommendations have been instrumental in significantly enhancing the scientific rigor and clarity of presentation of our study.

In response to all the reviewers' comments, we have undertaken a comprehensive and meticulous revision of the original manuscript. The primary improvements include:

1. Comprehensive Strengthening of Statistical Methods: Following the recommendations, we used binary logistic regression for clinical data analysis and introduced baseline characteristics comparison.

2. In-Depth Elucidation of Clinical Significance: We have tempered the language regarding conclusions on "long-term outcome improvement" to reflect a more cautious interpretation.

3. Clarification of Limitations and Future Directions: A new paragraph has been added to the Discussion section, openly addressing the study's limitations (e.g., sample size, single-center design). Based on the reviewers' suggestions, we have also outlined a clear pathway for future prospective randomized controlled trials.

4. Financial disclosure has been changed: this research was funded by Tianjin Municipal Education Commission granted number 2024KJ248.

We firmly believe that these revisions have substantially elevated the methodological robustness, reliability of the results, and overall persuasiveness of the conclusions.

We kindly request that you consider our revised manuscript for the next stage of the review process and await your further decision.

---

## [Editor Report · Decision Letter 2]

24 Nov 2025

Dear Dr. Mahamat,

Thank you for submitting your manuscript to PLOS ONE. After careful consideration, we feel that it has merit but does not fully meet PLOS ONE’s publication criteria as it currently stands. Therefore, we invite you to submit a revised version of the manuscript that addresses the points raised during the review process.

**Thank you for responding so quickly with the response to reviewers formatted as instructed. Unfortunately, one of the major critiques has still not been addressed. Reviewer 1 makes the point in question/comment 3 that removing patients who had post-operative survival < 6 months invalidates the study, and I agree with this point. In a clinical study describing a novel therapy, It is reasonable to remove patients who are lost to follow up. However, it is not reasonable to remove patients who die shortly following treatment. Such actions will clearly skew the results. As such, please re-incorporate this data and add back any of the patients who died less than 6 months after aSAH that you had removed. I look forward to seeing your revision.**

**Best regards, **
**Eric**

We look forward to receiving your revised manuscript.

Kind regards,

Eric Anthony Sribnick, MD, PhD, FAANS

Academic Editor

PLOS ONE

Journal Requirements:

Additional Editor Comments:

Thank you for responding so quickly with the response to reviewers formatted as instructed. Unfortunately, one of the major critiques has still not been addressed. Reviewer 1 makes the point in question/comment 3 that removing patients who had post-operative survival < 6 months invalidates the study, and I agree with this point. In a clinical study describing a novel therapy, It is reasonable to remove patients who are lost to follow up. However, it is not reasonable to remove patients who die shortly following treatment. Such actions will clearly skew the results. As such, please re-incorporate this data and add back any of the patients who died less than 6 months after aSAH that you had removed. I look forward to seeing your revision.

Best regards,

Eric

---

## [Author Response · Author response to Decision Letter 3]

10 Dec 2025

We sincerely thank the reviewer for this critical and insightful comment.We have re-integrated all patients with aSAH who died within 6 months of treatment and were previously excluded from the primary functional outcome analysis. The dataset has been updated accordingly.All relevant sections in the manuscript (Methods, Results, Tables, and Figures) have been thoroughly updated to reflect the new, complete cohort and the results from the revised statistical models.

---

## [Editor Report · Decision Letter 3]

15 Dec 2025

Comparative Efficacy of Cisternal Drainage versus External Ventricular Drainage in Severe Aneurysmal Subarachnoid Hemorrhage

PONE-D-25-33307R3

Dear Dr. Mahamat,

We’re pleased to inform you that your manuscript has been judged scientifically suitable for publication and will be formally accepted for publication once it meets all outstanding technical requirements.

Kind regards,

Eric Anthony Sribnick, MD, PhD, FAANS

Academic Editor

PLOS One

Additional Editor Comments (optional):

The response to reviewers' and editor comments is now complete.
---

## [Editor Report · Acceptance letter]

PONE-D-25-33307R3

PLOS One

Dear Dr. Mahamat,

I'm pleased to inform you that your manuscript has been deemed suitable for publication in PLOS One. Congratulations! Your manuscript is now being handed over to our production team.

Kind regards,

on behalf of

Dr. Eric Anthony Sribnick

Academic Editor

PLOS One